# The Efficacy and Safety of EUS-Guided Gallbladder Drainage as a Bridge to Surgery for Patients with Acute Cholecystitis

**DOI:** 10.3390/jcm12082778

**Published:** 2023-04-08

**Authors:** Ken Ishii, Yuji Fujita, Eisuke Suzuki, Yuji Koyama, Seitaro Tsujino, Atsuki Nagao, Kunihiro Hosono, Takuma Teratani, Kensuke Kubota, Atsushi Nakajima

**Affiliations:** 1Department of Hepato-Biliary-Pancreatic Medicine, NTT Tokyo Medical Center, 5-9-22 Higashi-Gotanda, Shinagawa-ku, Tokyo 141-8625, Japan; meitokuso@yahoo.co.jp (K.I.);; 2Department of Surgery, NTT Tokyo Medical Center, Tokyo 141-8625, Japan; 3Department of Gastroenterology and Hepatology, Yokohama City University School of Medicine, Yokohama 236-0004, Japan

**Keywords:** EUS drainage, acute cholecystitis, bridge to surgery

## Abstract

Background and Aim: This study aimed to compare the efficacy and safety of endoscopic ultrasound-guided gallbladder drainage and percutaneous transhepatic gallbladder drainage as a bridge to surgery in patients with acute cholecystitis unfit for urgent cholecystectomy. Methods: This retrospective study included 46 patients who underwent cholecystectomy following endoscopic ultrasound-guided gallbladder drainage (EUS-GBD) or percutaneous transhepatic gallbladder drainage (PTGBD) for acute cholecystitis in NTT Tokyo Medical Center. We surveyed 35 patients as the EUS-GBD group and 11 patients as the PTGBD group, and compared the rate of technical success of the cholecystectomy and periprocedural adverse events. A 7-F, 10-cm double pigtail plastic stent was used for ultrasound-guided gallbladder drainage. Results: The rate of technical success of cholecystectomy was 100% in both groups. Regarding postsurgical adverse events, no significant difference was noted between the two groups (EUS-GBD group, 11.4%, vs. PTGBD group, 9.0%; *p =* 0.472). Conclusions: EUS-GBD as a BTS seems to be an alternative for patients with AC because it can ensure lower adverse events. On the other hand, there are two major limitations in this study––the sample size is small and there is a risk of selection bias.

## 1. Introduction

Cholecystectomy is the curative treatment for acute cholecystitis (AC). Early cholecystectomy is mandatory for AC; however, emergency cholecystectomy for AC is associated with high morbidity (20–30%) and mortality (6–30%) rates in patients with significant comorbidities [1,2,3]. As a result, some surgeons prefer non-surgical procedures as makeshift treatments, such as antibiotic administration with/without percutaneous/endoscopic drainage, as an alternative to emergent cholecystectomy. However, elective surgery may lead to several complications, including empyema, gangrene, perforation, pericholecystitis with abscess formation, peritonitis, and sepsis [4,5]. Emergent surgery may not be safe and practical in patients with high surgical risk [6]. Percutaneous transhepatic gallbladder drainage (PTGBD) has been performed as a bridge for delayed surgical treatment in vulnerable patients with high surgical risk. The presence of a drainage tube may increase the risk of an adverse event during surgery by 16.2% to 25% [7,8,9]. Recently, endoscopic ultrasound-guided gallbladder drainage (EUS-GBD) has gained attention as a treatment for internal drainage of the gallbladder in high-risk patients [10,11,12,13,14,15,16,17,18]. Although PTGBD followed by late laparoscopic cholecystectomy for high-risk patients has been accepted as the standard procedure [19,20,21], there are limitations of PTGBD, such as inconvenience for patients and risk of its dislocation. However, there are no reports on alternatives to PTGBD focusing on the bridge to surgery (BTS). Thus, the present study aimed to validate the efficacy and safety of EUS-GBD as a BTS in patients with AC who are considered unfit for urgent cholecystectomy.

## 2. Materials and Methods

### 2.1. Study Design

This was a retrospective study conducted between April 2016 and July 2021. This study protocol was approved by the Institutional Review Board (ID18-313) of our institute. The study was investigator-initiated and conducted according to the ethical principles of the Declaration of Helsinki. Written informed consent was obtained from all patients.

### 2.2. Patients

Patients with a diagnosis of AC admitted to our institute between April 2016 and July 2021 were retrospectively identified. The diagnosis of AC was made using a combination of patient history, physical examination, laboratory analysis, and imaging (abdominal ultrasonography, computed tomography, and magnetic resonance imaging), and based on the Tokyo Guideline 2018 [22]. Patients with common bile duct stones were excluded because they had concurrent cholangitis. Patients were divided into two groups: one group who underwent cholecystectomy following EUS-GBD during the period from April 2019 to June 2021, and another group who underwent cholecystectomy following PTGBD during the period from April 2016 to June 2018.

### 2.3. Procedures

#### 2.3.1. EUS-GBD

EUS-GBD was performed by endoscopists who had performed over 500 interventional-EUS procedures and over 500 therapeutic ERCP procedures. Endoscopists used an oblique-viewing, curved-linear array echoendoscope (GF-UCT260 or GF-UCT240; Olympus Medical Systems, Tokyo, Japan) and a dedicated processor (ME-1/2; Olympus Medical Systems). The gallbladder was depicted by ultrasound imaging from the duodenal bulb or gastric antrum and punctured using a 19-gauge fine aspiration needle (EZ Shot 3 Plus; Olympus Medical Systems). Thereafter, a 0.025-inch guidewire (VisiGlide2; Olympus Medical Systems) was inserted into the gallbladder lumen, and the tract was dilated using a 4-mm balloon catheter with a tapered tip (REN; Kaneka Corporation, Tokyo, Japan). Finally, a 7-Fr 10-cm double-pigtail plastic stent (DPPS) (Through & Pass DP; Gadelius Medical K.K, Tokyo, Japan) was placed in the gallbladder through the duodenal bulb or gastric antrum (Figure 1). The inclusion criteria were: obvious cholecystitis identified by the presence of gallstones, no gallbladder perforation, and the provision of written informed consent.

#### 2.3.2. PTGBD

PTGBD was performed under local anesthesia by trained interventional radiologists in the interventional suite. A transhepatic route was used in all patients, and a 7-Fr pigtail drainage catheter (Hanako Medical Co., Ltd., Saitama, Japan) was placed between the seventh or eighth intercostal space under combined sonographic and fluoroscopic guidance.

### 2.4. Follow-Up

All patients underwent plain abdominal radiography and laboratory tests the day after the procedure and leading up to the surgery. Oral diet was started when clinical symptoms improved without any severe adverse events. DPPS was kept in place without periodical exchange until the surgery.

### 2.5. Laparoscopic Cholecystectomy

Laparoscopic cholecystectomy was performed for eligible patients at least 1 month after EUS-GBD. The previous day before the surgery, the DPPS was endoscopically removed. The surgery was performed under general anesthesia using a standard four-trocar technique. Surgeons identified the enterocholecysto fistula, which was then immediately cut using a stapler. If the laparoscopic surgery was difficult to complete, conversion to open cholecystectomy was performed at the operator’s discretion. All laparoscopic cholecystectomy procedures were performed by one hepatobiliary pancreatic surgeon who had previously performed more than 500 laparoscopic cholecystectomies.

Difficult laparoscopic cholecystectomy (DLC) was defined as a procedure with an operative time ≥ 3 h, bleeding volume ≥ 300 mL common bile duct injury, partial cholecystectomy, the need for a second surgeon, and/or conversion to open surgery [22].

### 2.6. Outcomes

The primary outcome was technical success of the cholecystectomy after EUS-GBD. Technical success was defined as successful gallbladder removal during cholecystectomy without complications. Clinical success was defined as clinical improvement (resolution of fever, decrease in white blood cell count, and resolution of pain and tenderness) within 72 h after the procedure. The secondary outcome was periprocedural adverse events including prolonged surgical time after cholecystectomy.

### 2.7. Statistical Analysis

Data are summarized as mean ± standard deviation for continuous data and as frequency and percentages for categorical data. For continuous data, characteristics and outcomes of the two groups were compared using the student’s *t*-test or Mann–Whitney U test based on the viability of the normality assumption. The Chi-squared or Fisher’s exact test was used to compare the two groups with regard to categorical characteristics and outcomes. The level of significance was set at a two-sided *p*-value < 0.05. Statistical analysis was performed using BellCurve for Excel (Social Survey Research Information Co., Ltd., Tokyo, Japan).

## 3. Results

### 3.1. Patient Characteristics

In this period, 46 patients were included in this study (Figure 1): 35 patients underwent EUS-GBD (62.9% male; average age, 69.2 ± 13.4 years) and 11 patients underwent PTGBD (90.9% male; average age, 72.4 ± 12.2 years), followed by cholecystectomy. No statistical differences were found in age, sex, or body mass index between the two groups (Table 1). The etiology of cholecystitis was gallstone disease (*n* = 35, 100%) in the EUS-GBD group and gallstone disease (*n* = 10, 81.8%), acalculous disease (*n* = 1, 9.1%), and gallbladder cancer (*n* = 1, 9.1%) in the PTGBD group (*p* = 0.005). No significant differences were noted regarding baseline diseases, advanced cancers (*p* = 0.721), cerebrovascular disorder (*p* = 0.912), or cardiopulmonary disease (*p* = 0.886) between the two groups. The severities for cholecystitis were moderate (*n* = 33, 94.3%) and severe (*n* = 2, 5.7%) in the EUS-GBD group and moderate (*n* = 11, 100%) in the PTGBD group. Cholecystectomy was proposed for all patients at the initial diagnosis for AC; however, if the surgeons, endoscopists, and radiologists regarded these patients as unsuitable surgical candidates, either EUS-GBD or PTGBD was performed.

### 3.2. Primary Outcome

Clinical success of gallbladder drainage was achieved in 100% of patients in the EUS-GBD group and 81.8% of patients in the PTGBD group; two patients in the PTGBD group exhibited catheter dislodgement. No significant difference was observed regarding the duration from drainage to cholecystectomy between the two groups (*p* = 0.512).

Technical success of cholecystectomy was achieved in 100% of patients in both groups (Table 2). All patients in the EUS-GBD group underwent laparoscopic cholecystectomy, and only one (2.9%) patient required conversion to open surgery. In the PTGBD group, eight patients (72.7%) underwent laparoscopic cholecystectomy, three patients (27.3%) underwent open cholecystectomy, and one patient (12.5%) required conversion to open cholecystectomy. The number of patients who required conversion was not statistically different between the two groups (*p* = 0.400). No significant differences were noted regarding operation time (*p* = 0.707), estimated blood loss (*p* = 0.493), or duration from operation to discharge (*p* = 0.541) between the two groups.

### 3.3. Secondary Outcome

Postsurgical adverse events were observed in four patients (11.4%) in the EUS-GBD group and in one patient (9.0%) in the PTGBD group; no significant differences were found between the two groups (*p =* 0.472) (Table 3). In the EUS-GBD group, four patients suffered from abscesses that were managed by adjusting the position of the drain placed at the time of cholecystectomy. In the PTGBD group, the single adverse event was postoperative heart failure, managed with medication.

## 4. Discussion

This paper indicated that EUS-GBD could be an alternative to PTGBD as a BTS. Ryu’s meta-analysis and systematic review reported EUS-GBD was comparable with PTGBD regarding clinical success, with less reintervention and readmission, for acute cholecystitis with high surgical risk [23]. However, postprocedural adverse events, which could be conservatively managed, occurred in 6 of 35 patients (17.1%) in the EUS-GBD group; controllable peritonitis occurred in all patients. As bile leak reportedly occurs in one in eight (12.5%) patients with DPPS [24], the rate of bile leak in this study (17.1%) was relatively high. Although a 4-mm balloon catheter was used in all patients in our study, a high rate of bile leak may have occurred due to the use of this catheter, and leakage after the dilation procedure was convertibly countered. On the other hand, postprocedural adverse events occurred in 3 of 11 patients (27.2%) in the PTGBD group, and 2 of these patients exhibited drain dislodging. Bile leak peritonitis can be treated conservatively with antibiotics, but drain dislodging is a serious adverse event. This suggests that EUS-GBD is an acceptable method for BTS in terms of adverse events. In the PTGBD group, one patient had gallbladder cancer as the etiology of acute cholecystitis. The length of time from operating to discharge for this patient was 5 days. In addition, this patient did not require conversion to open. Therefore, in our report, gallbladder cancer was not affected by length of time from operating to discharge and conversion to open.

Moreover, concerning difficult LC (DLC), the rate of DLCs was relatively high compared to a previous paper [25] (45.7% vs. 26.3%). In other reports, 3 of 12 (25%) patients and 2 of 23 (9%) patients required conversion to open cholecystectomy [26,27]; in our study, only 1 of 35 (2.9%) patients required conversion. Thus, LC led EUS-GBD could be endured when it comes to patients with DLC.

Jang et al. [27] reported rates of conversion to open cholecystectomy after EUS-GBD had an adverse effect on laparoscopic cholecystectomy and showed that EUS-GBD did not cause severe inflammation or adhesion to surrounding gallbladder tissue; however, this study only included surgical candidates, and cholecystectomy was performed after a median of 5 days after EUS-GBD. In our study, cholecystectomy was performed as elective surgery based on the results of Altieri et al. [28], who revealed that a duration of ≤8 weeks (*n* = 1211) was associated with a higher overall rate of complications.

A well-timed LC 8 weeks after EUS-GBD would be preferable, since the inflammation would be ameliorated, ensuring better surgical outcomes [28]. In our study, the duration from drainage to cholecystectomy in the EUS-GBD group was 86.7 days; this was >8 weeks and longer than the duration in the PTGBD group. However, in the report by Altieri et al., the average time to cholecystectomy was 203 days in the >8 weeks group [28]. Therefore, the rate of DLC in our study could be lower if the waiting period for the surgery was lowered. All patients in our study demonstrated moderate or mild adhesions and fibrosis during surgery; nevertheless, surgery was performed safely, and despite the presence of adhesions and fibrosis, only one patient required conversion to open cholecystectomy. This also indicated that the inflammation due to EUS-GBD can be a surmountable event for experienced laparoscopic surgeons. The EUS-GBD group showed moderate and mild adhesions and fibrosis in all of the patients, yet despite these adhesions and fibrosis, as far as we can observe, there are no long-term postoperative complications such as upper gastrointestinal obstruction in the two groups.

Adverse events (AEs) due to drainage present an independent risk for postsurgical adverse events. In our study, peritonitis and drain dislodging were the most common postprocedural AEs, with bile leak closely related to these events. Bile leak may make cholecystectomy difficult due to the severe adhesion around the gallbladder and enterocholecysto fistula; thus, to minimize the risk of bile leak in EUS-GBD, lumen-apposing metal stent (LAMS) is used. EUS-GBD using LAMS is becoming a widely accepted therapeutic approach for gallbladder drainage with high clinical and technical success rates and low rates of adverse events, as shown by several studies [29]; however, it is only covered by insurance for pancreatic pseudocyst and walled-off necrosis in Japan. Therefore, although plastic stents were used in the EUS-GBD group in our study, it may be that LAMS provides more safety during the procedure [29].

In one previous report, AC had clinical particularities in aged patients with an increased rate of postoperative complications [30]. We obtained the same result in our study. In an aging society, PTGBD is a routine procedure; however, dislocation would be critical for patients with AC. Indeed, drain migration is reported in 0.3–12% of patients [1,31,32,33,34]; besides, EUS-GBD in our study resulted in few cases of drain migration. Therefore, EUS-GBD will be safer and more reliable in the future. EUS-GBD would be more patient-friendly than the PTGBD without dislocation and inconvenience.

A review conducted by Lee et al. [35] revealed that nine patients demonstrated rapid clinical improvement within 72 h after EUS-GBD. Elective laparoscopic cholecystectomy was eventually performed in seven patients and was successful in six patients, and transduodenal cholecystostomy was converted to open cholecystectomy in one patient (14.3%) without complication. The rate of technical success of cholecystectomy was 100% in the report of both Lee et al. and our own report, whereas the rates of conversion to open cholecystectomy were 14.3% and 2.9%; thus, both studies demonstrate that LC following EUS-GBD was safe.

This study had some limitations. First, this was a retrospective study. Doctors’ treatment preferences may have resulted in a bias. The decision to PTGBD or EUS drainage was made at the discretion of the surgeons, endoscopists, and radiologists, and it may have led to a selection bias. Furthermore, due to the characteristics of our hospital in this study, the proportion of patients with underlying medical conditions was high, so the population may be slightly different from the usual acute cholecystitis patients. This may limit the generalizability of the study.

Second, the sample size of PTGBD patients included was small. In Ryu’s meta-analysis and systematic review, reported EUS-GBD was associated with fewer adverse events than PTGBD [23]. However, in our study, post procedural adverse events were observed in six patients (17.1%) in the EUS-GBD group and in three patients (27.2%) in the PTGBD group; no significant differences were found between the two groups (*p* = 0.472). However, no significant difference is seen, although that does not mean there are no differences between EUS-GBD and PTGBD. Hence, randomized controlled trials or non-inferiority trials with more patients should be planned to prove the present results.

Third, our study was conducted by only one expert hepatobiliary pancreatic surgeon; therefore, it may not be valid to generalize our results across other centers, as the surgeons may have varying levels of clinical experience and familiarity with cholecystectomy for high-risk patients with acute cholecystitis. Hence, larger prospective studies are required to confirm our results. Third, since LAMS cannot be used for EUS-GBD in Japan, we hope that a global study using LAMS will be conducted in the future.

In conclusion, this paper indicated that EUS-GBD could be an alternative to PTGBD as a BTS. However, further studies are needed to confirm this.

## Figures and Tables

**Figure 1 jcm-12-02778-f001:**
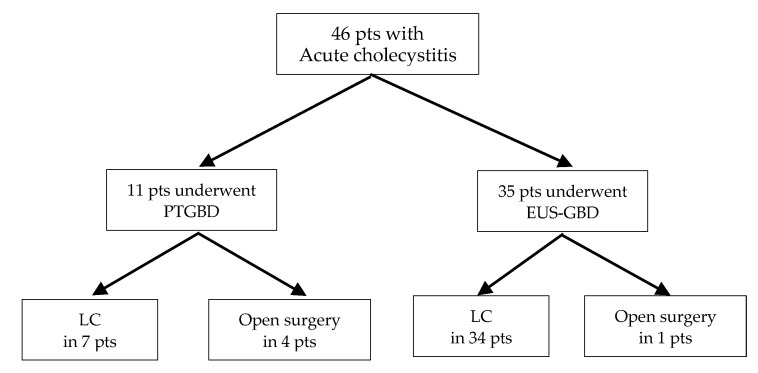
Result of analyzed patients (pts) with acute cholecystitis.

**Table 1 jcm-12-02778-t001:** Patient characteristics.

Variable	EUS-GBD (*n* = 35)	PTGBD (*n* = 11)	*p* Value
Age (years)	69.2 ± 13.4 (34–88)	72.4 ± 12.2 (44–82)	0.900
Sex (male/female)	22/13	10/1	0.052
BMI	24.2 ± 3.8 (15–32.8)	22.9 ± 2.6 (18.9–25.8)	0.286
Etiology of cholecystitis			
Gallstone	35 (100)	9 (81.8)	0.005
Acalculous	0	1 (9.0)	
Gallbladder cancer	0	1 (9.0)	
Underlying conditions			
Baseline disease			
Advanced cancer	6 (17.1)	3 (27.3)	0.721
Cerebrovascular disorder	2 (5.7)	1 (9.1)	0.912
Cardiopulmonary disease	8 (22.9)	2 (18.2)	0.886
ASA-PS I	5 (14.3)	1 (9.0)	0.445
ASA-PS II	28 (80.0)	6 (54.5)	0.094
ASA-PS III	2 (57.1)	3 (27.3)	0.272
ASA-PS IV	0	1 (9.0)	
Severity of cholecystitis (based on Tokyo guideline 2018)			
Moderate	33 (94.3)	11 (100)	0.201
Severe	2 (5.7)	0	

Numbers are shown in number (%) or average ± SD (range); EUS-GBD, endoscopic ultrasound-guided gallbladder drainage; PTGBD, percutaneous transhepatic gallbladder drainage; BMI, body mass index; ASA-PS, American Society of Anesthesiologists physical status; SD, standard deviation.

**Table 2 jcm-12-02778-t002:** Comparison of drainage procedure outcomes.

Variable	EUS-GBD (*n* = 35)	PTGBD (*n* = 11)	*p* Value
Technical success of gallbladder drainage	35 (100)	11 (100)	
Clinical success of gallbladder drainage	35 (100)	9 (81.8)	0.005
Procedure time (min)	25.1 ± 9.2 (13–52)	No record	
Time from drainage to cholecystectomy (days)	86.7 ± 113.7 (29–632)	62.0 ± 87.8 (7–308)	0.512
Technical success of cholecystectomy	35 (100)	11 (100)	
Type of cholecystectomy			
Laparoscopic	35 (100)	8 (72.7)	0.002
Open	0	3 (27.3)	
Laparoscopic converted to open	1 (2.9)	1 (12.5)	0.4
Operating time (min)	171.9 ± 71.7 (46–368)	182.0 ± 53.5 (110–302)	0.707
Estimated blood loss (ml)	75.5 ± 99.5 (5–400)	103.2 ± 130.8 (10–440)	0.493
Time from operation to discharge (days)	5.4 ± 2.5 (3–14)	6.5 ± 2.8 (3–13)	0.541

Numbers are shown as number (%) or average ± SD (range); EUS-GBD, endoscopic ultrasound-guided gallbladder drainage; PTGBD, percutaneous transhepatic gallbladder drainage; SD, standard deviation.

**Table 3 jcm-12-02778-t003:** Comparison of adverse events.

Variable	EUS-GBD (*n* = 35)	PTGBD (*n* = 11)	*p* Value
Post procedural adverse events	6 (17.1)	3 (27.2)	0.361
Types of adverse events			
Recurrent cholecystitis	0	1 (9.0)	0.035
Drain dislodging	0	2 (18.2)	0.005
Peritonitis	6 (17.1)	0	0.071
Patients requiring repeat procedure	0	0	
Postsurgical adverse events	4 (11.4)	1 (9.0)	0.472
Recurrent biliary events	0	0	
Abscess	4 (11.4)	0	0.418

Numbers are shown as number (%) or average ± SD (range); EUS-GBD, endoscopic ultrasound-guided gallbladder drainage; PTGBD, percutaneous transhepatic gallbladder drainage; SD, standard deviation.

## Data Availability

Not applicable.

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
