# Peer review of "The Efficacy and Safety of EUS-Guided Gallbladder Drainage as a Bridge to Surgery for Patients with Acute Cholecystitis"

_jcm, 2023, doi:10.3390/jcm12082778_

Round 1

Reviewer 1 Report (New Reviewer)

Thank you for the opportunity to review the manuscript titled: “The efficacy and safety of EUS-guided gallbladder drainage as a bridge to surgery for patients with acute cholecystitis” by Ishii et al. In this retrospective study 11 patients that underwent percutaneous gallbladder drainage were compared to 35 patients undergoing EUS-guided gallbladder drainage. The number of patients in this retrospective study is small. No significant differences were seen between the two groups. This leads to several issues:

1.       If no difference is seen, that does not mean there are no differences. The number of patients is too small to draw any solid conclusions.

2.       The high proportion of patients with underlying conditions such as advances cancer makes it a highly heterogeneous population in both groups. This limits the generalizability of the study.

3.       The authors conclude that EUS-GBD can ensure lower adverse events. This cannot be based on the results of the current study.

Author Response

  1. If no difference is seen, that does not mean there are no differences. The number of patients is too small to draw any solid conclusions.

-Response: The authors would like to thank the reviewer for his/her constructive critique to improve the manuscript. We have made every effort to address the issues raised and to respond to three comments. Please, find next a detailed, point-by-point response to the reviewer's comments. We hope that our revisions will meet the reviewer’s expectations.

Certainly, the number of patients in the study was small. And, as you pointed out, no significant difference is seen, that does not mean there are no differences between EUS-GBD and PTGBD. So, to support the results of this study. We have mentioned randomized controlled trials or non-inferiority trial with more patients should be planned to prove the present results at limitation section. (Line257-264)

  1. The high proportion of patients with underlying conditions such as advances cancer makes it a highly heterogeneous population in both groups. This limits the generalizability of the study.

-Response: We would like to thank the editor for the suggestion. As you pointed out, the background disease is different from the general group of patients with acute cholecystitis, but in this study, there was no significant difference between the two groups.  We regarded “No significant differences were noted regarding baseline diseases” in Line 129-130. To communicate more accurately to the reader, we have added “due to the characteristics of our hospital in this study, the proportion of patients with underlying medical conditions was high, so the population may be slightly different from the usual acute cholecystitis patients. This may limit the generalizability of the study.” in Line253-256.

  1. The authors conclude that EUS-GBD can ensure lower adverse events. This cannot be based on the results of the current study.

-Response: We would like to thank the editor for the suggestion. As you pointed out, both EUS-GBD, PTGBD have adverse events. So, we deleted "EUS-GBD assured LC and reduced drainage-related complications" and "This suggests that EUS-GBD is an acceptable method of BTS which is associated with less complications and a higher rate of LC than its counterpart" at discussion section. Furthermore, we're trying not to mislead our readers, we have paraphrased our conclusion. (Line250-272)

Reviewer 2 Report (Previous Reviewer 2)

The authors have addressed the issues raised by this reviewer, no more questions.

Author Response

Thank you

Reviewer 3 Report (Previous Reviewer 1)

Thanks to the authors for the careful revision.

The impact of patient co-morbidity, the gallbladder cancer status in the PTGBD group on the research, and the long-term complications of adhesions and fibrosis in the EUS-GBD group have been detailed point-by-point in the article by the authors. In addition, the small sample size of patients in the PTGBD group could not be effectively corrected due to the retrospective nature of this research, but the authors have also explained this in the article and used more conservative and precise wording in the discussion. Overall, the latest manuscript is an improvement over the previous manuscript.

Author Response

Thank you

Round 2

Reviewer 1 Report (New Reviewer)

The authors improved the manuscript by taking the issues I mentioned into account. 

This manuscript is a resubmission of an earlier submission. The following is a list of the peer review reports and author responses from that submission.

Round 1

Reviewer 1 Report

In that retrospective study, the authors tried to demonstrate that EUS-GBD would be better than PTGBD as a BTS for patients with moderate or severe acute cholecystitis based on the Tokyo guideline 2018.

Some concerns:

1.Patient co-morbidities were not specified, e.g., were the 46 patients previously healthy? Are there any basic diseases or genetic disorders? All of these may affect the outcome of the procedure and the surgery.

2.The sample size of PTGBD patients included was too small (11 cases), resulting in an impact on statistical efficacy when comparing them to the EUS-GBD group. It does not adequately account for the fact that this BTS has fewer complications relative to conventional PTGBD.

3.P-values for the two groups showed a difference in the etiology of cholecystitis. patients in the PTGBD group had 1 patient with gallbladder cancer, does this significantly increase the length of stay after the operation in the PTGBD group as well as the number of open surgeries statistically?

4.The EUS-GBD group showed moderate and mild adhesions and fibrosis in all of patients. Do these adhesions and fibrosis increase the incidence of long-term postoperative complications such as upper gastrointestinal obstruction compared to the PTGBD group? This is not stated in the text.

Author Response

January 16, 2023

Dear Phoebe Zhang

 Journal of Clinical Medicine

Dear Editor:

We wish to re-submit the attached manuscript entitled “The efficacy and safety of EUS-guided gallbladder drainage as a bridge to surgery” The manuscript has been rechecked and appropriate changes have been made in accordance with the reviewers’ suggestions. The responses to their comments have been prepared and attached herewith.

We thank you and the reviewers for your thoughtful suggestions and insights, which have enriched the manuscript and produced a better and more balanced account of the research. We hope that the revised manuscript is now suitable for publication in your journal.

Thank you for your consideration. We look forward to hearing from you.

Sincerely,

Yuji Fujita

Department of Hepato-Biliary-Pancreatic Medicine, NTT Tokyo Medical Center, 5-9-22 Higashi-Gotanda, Shinagawa-ku, Tokyo, Japan

Telephone: +81-3448-6111

Fax: 81-3448-6071

Email: yufuji5395@gmail.com

Reviewer 1

1.Patient co-morbidities were not specified, e.g., were the 46 patients previously healthy? Are there any basic diseases or genetic disorders? All of these may affect the outcome of the procedure and the surgery.

Response: The authors would like to thank the reviewer for his/her constructive critique to improve the manuscript. We have made every effort to address the issues raised and to respond to all comments. Please, find next a detailed, point-by-point response to the reviewer's comments. We hope that our revisions will meet the reviewer’s expectations.

We have added basic diseases of the patients in the Results (Lines 130-132), and in the Table1. Diseases were divided into three categories, Advanced cancer, Cerebrovascular disorder, and Cardiopulmonary disease. The classification of diseases was based on this paper “EUS-guided cholecystoduodenostomy for acute cholecystitis with an anti-stent migration and anti-food impaction system; a pilot study” Ther Adv Gastroenterol 2016, Vol. 9(1) 19–25.

2.The sample size of PTGBD patients included was too small (11 cases), resulting in an impact on statistical efficacy when comparing them to the EUS-GBD group. It does not adequately account for the fact that this BTS has fewer complications relative to conventional PTGBD.

Response: We would like to thank the reviewer for the suggestion. We have added the explanation of small of the sample size in the Discussion as follows:

“And the sample size of PTGBD patients included was small.” (Lines 266–267)

In addition, the conclusion section in the Discussion has been changed as follows:

“EUS-GBD as a BTS seems to be an alternative for patients with AC because it can ensure lower adverse events, on the other hand, additional work is needed to confirm this.” (Lines 273–275)

  1. P-values for the two groups showed a difference in the etiology of cholecystitis. patients in the PTGBD group had 1 patient with gallbladder cancer, does this significantly increase the length of stay after the operation in the PTGBD group as well as the number of open surgeries statistically?

Response: We would like to thank the reviewer for the comment. Please note that we have examined the etiology of cholecystitis and added them to the Discussion section as follows. Result of the verification, we figured gallbladder cancer had no effect on anything.

“In the PTGBD group, one patient had gallbladder cancer as the etiology of acute cholecystitis. The length of hospital stay post procedure and time from operating to discharge for this patient were 16 days and 5 days. Besides, this patient did not require conversion to open. Therefore, in our report, gallbladder cancer was not affected length of hospital stay post procedure, time from operating to discharge and conversion to open.” (Lines 202-207)

  1. The EUS-GBD group showed moderate and mild adhesions and fibrosis in all of patients. Do these adhesions and fibrosis increase the incidence of long-term postoperative complications such as upper gastrointestinal obstruction compared to the PTGBD group? This is not stated in the text.

Response: We would like to thank the reviewer for the suggestion. As per the reviewer’s insightful suggestion, we have reviewed the data and realized that we had no long-term postoperative complications due to adhesions and fibrosis in all patients. We have added the explanation of no long-term in the postoperative complications due to adhesions and fibrosis in the Discussion as follows:

“The EUS-GBD group showed moderate and mild adhesions and fibrosis in all of patients, however, despite these adhesions and fibrosis, as far as we can observe there is no long-term postoperative complications such as upper gastrointestinal obstruction in the two group.” (Lines 231-234)

In addition to the point you mentioned, there was an error in the numerical value, which has been corrected. In the figure1, the numbers of patients in the PTGBD group was incorrected. There were 11 patients who underwent PTGBD, 7 patients who underwent LC and 4 patients who underwent open surgery.

Reviewer 2 Report

In the manuscript submitted by Ishii et al, the authors compared the efficacy and safety of endoscopic ultrasound-guided gallbladder drainage (EUSGBD) to percutaneous transhepatic gallbladder drainage (PTGBD) as a bridge to surgery for patients with acute cholecystitis but unfit for urgent cholecystectomy. Comparisons between the two procedures for clinical success and adverse events have been conducted by many researchers, and several meta-analysis studies also revealed that EUSGBD was comparable with PTGBD regarding clinical success, with less reintervention and readmission for acute cholecystitis with high surgical risk. Furthermore, the cholecystitis recurrence rate was lower with EUSGBD with LAMS. However, as the authors pointed out, no comparison had been performed for EUSGBD as a bridge-to-surgery for late laparoscopic cholecystectomy.  Therefore, the topic of the current study was of interest, and the results provided some preliminary basis for the application of this alternative approach. There are some minor issues should be addressed:

1.       References for the comparison between the two approaches should be added, particularly those meta-analysis studies.

2.       In the limitation part, the small patient size should be stressed.

3.       Line 209, “Altieri et al”, the reference was not in the right format.

4.       Minor grammar errors/typos should be corrected.   

Author Response

January 16, 2023

Dear Phoebe Zhang

 Journal of Clinical Medicine

Dear Editor:

We wish to re-submit the attached manuscript entitled “The efficacy and safety of EUS-guided gallbladder drainage as a bridge to surgery” The manuscript has been rechecked and appropriate changes have been made in accordance with the reviewers’ suggestions. The responses to their comments have been prepared and attached herewith.

We thank you and the reviewers for your thoughtful suggestions and insights, which have enriched the manuscript and produced a better and more balanced account of the research. We hope that the revised manuscript is now suitable for publication in your journal.

Thank you for your consideration. We look forward to hearing from you.

Sincerely,

Yuji Fujita

Department of Hepato-Biliary-Pancreatic Medicine, NTT Tokyo Medical Center, 5-9-22 Higashi-Gotanda, Shinagawa-ku, Tokyo, Japan

Telephone: +81-3448-6111

Fax: 81-3448-6071

Email: yufuji5395@gmail.com

Reviewer 2

  1. References for the comparison between the two approaches should be added, particularly those meta-analysis studies.

Response: The authors would like to thank the reviewer for his/her constructive critique to improve the manuscript. We have made every effort to address the issues raised and to respond to all comments. Please, find next a detailed, point-by-point response to the reviewer’s comments. We hope that our revisions will meet the reviewer’s expectations.

Please note that we have examined meta-analysis studies and We have added the explanation in the Discussion as follows:

“Ryu's meta-analysis and systematic review reported EUS-GBD was comparable with PTGBD regarding clinical success, with less reintervention and readmission, for acute cholecystitis with high surgical risk.” (Lines 190–193)

  1. In the limitation part, the small patient size should be stressed.

Response: We would like to thank the reviewer for the suggestion. We have added the explanation of small of the sample size in the Discussion as follows:

“And the sample size of PTGBD patients included was small.” (Lines 266–267)

  1. Line 209, “Altieri et al”, the reference was not in the right format.

Response: We would like to thank the reviewer for the suggestion. This section has been redacted per the reviewer's suggestion.

  1. Minor grammar errors/typos should be corrected.   

Response: We would like to thank the reviewer for the suggestion. This text has been corrected by English proofreading.

In addition to the point you mentioned, there was an error in the numerical value, which has been corrected. In the figure1, the numbers of patients in the PTGBD group was incorrected. There were 11 patients who underwent PTGBD, 7 patients who underwent LC and 4 patients who underwent open surgery.